# The Significance of Coronary Artery Calcification for Percutaneous Coronary Interventions

**DOI:** 10.3390/healthcare12050520

**Published:** 2024-02-22

**Authors:** Paweł Lis, Marek Rajzer, Łukasz Klima

**Affiliations:** 1st Department of Cardiology, Interventional Electrocardiology and Arterial Hypertension, Jagiellonian University Medical College, 30-688 Kraków, Poland; plis@su.krakow.pl (P.L.); rajzer@su.krakow.pl (M.R.)

**Keywords:** coronary artery calcification, rotational atherectomy, orbital atherectomy, intravascular lithotripsy, intravascular ultrasound, optical coherence tomography

## Abstract

The prevalence of calcium deposits in coronary arteries grows with age. Risk factors include, e.g., diabetes and chronic kidney disease. There are several underlying pathophysiological mechanisms of calcium deposition. Severe calcification increases the complexity of percutaneous coronary interventions. Invasive techniques to modify the calcified atherosclerotic plaque before stenting have been developed over the last years. They include balloon- and non-balloon-based techniques. Rotational atherectomy has been the most common technique to treat calcified lesions but new techniques are emerging (orbital atherectomy, intravascular lithotripsy, laser atherectomy). The use of intravascular imaging (intravascular ultrasound and optical coherence tomography) is especially important during the procedures in order to choose the optimal strategy and to assess the final effect of the procedure. This review provides an overview of the role of coronary calcification for percutaneous coronary interventions.

## 1. Introduction

Coronary artery calcification (CAC) is a serious problem in the aging population. Percutaneous coronary interventions (PCI) in heavily calcified lesions pose a major challenge due to technical difficulties and possibly impair both short- and long-term outcomes. The presence of coronary artery calcification is age-dependent, thus, the number and complexity of PCI procedures grows with age. The accumulation of calcium in the arterial wall typically starts after the age of 40 and its prevalence gradually increases with age. CAC is present in 90% of men and 67% of women older than 70 [1]. There is some protective effect of estrogens observed both for the development of atherosclerosis in general as well as for the development of CAC, which is three times more frequent in postmenopausal females versus premenopausal females [2].

CAC formation begins as microcalcifications (0.5 to 15.0 μm) and then proceeds into larger forms of calcium fragments, which eventually lead to creation of sheet-like deposits (>3 mm) [3]. There are two main CAC types regarding the localization within the vessel wall, either within the intima or in the vascular medial layer. The latter type is more frequently observed in patients with peripheral artery disease and its progression is linked to renal failure, hypercalcemia, hyperphosphatemia, parathyroid hormone abnormalities, and duration of dialysis. This type of calcification is associated with arterial stiffness and increases the risk of cardiovascular events [4]. Intimal calcification, however, is a dominant pattern seen in coronary arteries and therefore is the main topic of this article.

In terms of pathophysiology, similar calcium regulatory mechanisms can affect both bone formation and growth as well as CAC progression [5]. Alkaline phosphatase is a potential molecular marker of vascular calcification and is crucial in early calcium deposition [6]. Vascular smooth muscle cells (VSMCs) produce matrix vesicles which play a crucial role in the regulation of mineralization in the vascular intima and media. In intimal calcification, the osteogenic differentiation of VSMCs is induced by inflammatory mediators and elevated lipid content within atherosclerotic lesions [7]. There are also other cells, including microvascular pericytes and adventitial myofibroblasts, that can generate matrix mineralization and differentiate into osteoblasts which results in calcified deposits.

Diabetes and chronic kidney disease are risk factors for the development of CAC. In diabetes, advanced end-products of glycation might promote the mineralization of microvascular pericytes [5]. The proper control of levels of glycemia can slow CAC progression in type 1 diabetes (but not type 2) [8,9]. Higher levels of glycated hemoglobin have been shown to be associated with the progression of CAC [3]. In chronic kidney disease, higher cardiovascular mortality and morbidity may be attributed mainly to accelerated atherosclerosis and the presence of CAC. Hypercalcemia and hyperphosphatemia lead to CAC progression. Phosphate affects calcium–phosphate solubility equilibrium, but also stimulates the osteochondrogenic transformation of VSMCs [10]. In patients with chronic kidney disease, heavily calcified lesions can develop at a younger age, as presented in the case of a 33-year-old patient in Figure 1.

The potential link between CAC and osteoporosis is the receptor activator of nuclear factor-kappaB ligand/osteoprotegerin pathway. The pro-osteoclastic and bone resorptive effects of the receptor activator of nuclear factor-kappa pathway are counterbalanced by osteoprotegerin which functions as a decoy receptor [6]. However, further research is needed in that matter because epidemiologic data in humans suggest that higher osteoprotegerin is associated with higher CAC incidence and more cardiovascular events [11].

There is no relationship between high calcium dietary intake and CAC progression. That fact suggests that CAC should be attributed not to simple calcium overload, but rather to disturbances in regulatory mechanisms [12].

The morphology of calcified lesions differs significantly patient-to-patient. Calcification is present in plaques of various types, such as fibrocalcific plaques, fibroatheromas, ruptured plaques, plaque erosions, etc. Microcalcification can be a factor making an unstable plaque more prone to rupture, such as in case of thin cap fibroatheroma (TCFA), in which microcalcification nodules act as local tissue stress concentrators [12]. We can speak of a so-called coronary calcium paradox. A high total calcification score is a marker of cardiovascular risk, but on the other hand, extensive local calcification with high density may be a marker of plaque stability [13]. There are two groups of patients with CAC: those with unstable plaques (often with spotty calcification), who are predominantly younger and are presenting with acute coronary syndromes, and patients with stable disease, predominantly older with heavily calcified plaques (extensive calcification), with a high Agatson score in computed tomography (CT), often with multivessel disease [3]. The fact that the presence of dense calcified plaque is associated with a lower risk of cardiovascular events when compared with the presence of calcified plaque with low CT density (independently of total CAC score) has been proved in the MESA (Multi-Ethnic Study of Atherosclerosis) study [14]. A key mechanism responsible for the phenomenon is the increase in calcified plaque density. Plaques that contain denser calcification are less prone to rupture, which results in a lower risk of coronary events compared with less calcified ones at any level of calcified plaque volume (CPV) [15]. In the PARADIGM (Progression of Atherosclerotic Plaque Determined by Computed Tomographic Angiography Imaging) registry over 900 patients were analyzed. Participants who were enrolled underwent clinically indicated serial coronary CT. In multivariable regression analyses, CPV was strongly associated with higher risk of incident major adverse cardiovascular events (MACE). However, the crucial finding of the study was the fact that percent CPV was remarkably protective after accounting for plaque volume. High percent CPV decreased the risk of plaque progression and MACE. Increasing percent CPV was a marker of plaque stability and reduced risk at both a lesion and patient level [16]. Interestingly, there are studies with use of intravascular ultrasound (IVUS) that have shown the regression of atherosclerotic plaque in patients treated with high-intensity statins [17,18]. An increased volume of calcified plaque and decreased volume of fibrous plaque and necrotic core were observed in those studies. That suggests that statins may increase the calcium content in the coronary atherosclerotic lesions [13].

CAC increases the risk of multiple complications of percutaneous coronary interventions (PCI), like stent underexpansion [19] (with subsequential stent thrombosis, in-stent restenosis), as well as vessel dissection or perforation. Dedicated lesion-preparation techniques are needed in many cases to facilitate stent implantation.

## 2. Diagnosis

### 2.1. Computed Tomography (CT)

CT is the most important noninvasive tool to assess coronary calcification. The amount of coronary artery calcium can be quantified using the Agatson score which is expressed as a numeric value ranging from 0 to several thousand. The Agatson score is a summed score of all calcified lesions that considers total calcified area and maximum density of calcification (>130 Hounsfield units [HU]) [3]. The score of 0 indicates the absence of detectable coronary artery calcium, while higher scores reflect increasing levels of calcification. An Agatson score of >400 indicates extensive calcification. A high calcium score is a risk factor for developing cardiovascular events. Even the presence of minimal CAC in CT (i.e., score 1–10) increases the risk of coronary artery disease, as compared to the absence of CAC [20]. Coronary CT potentially has an important role in the pre-procedural planning of PCI strategy (assessment of calcium and possibility of calculation of fractional flow reserve based on a CT scan) but is still underutilized in daily clinical practice [21].

Coronary CT scans are able to provide precise structural information regarding the vessel wall structure and can characterize the subtypes of atherosclerotic plaque. Some plaques that are not flow-limiting can still carry a high risk because of their certain histopathological features. Patients with such plaque types cannot be identified through coronary angiography and potentially can be overlooked. The functional assessment of the severity of stenosis that can be performed during angiography (fractional flow reserve) is also only a downstream surrogate marker of proximal stenosis and does not provide information regarding plaque composition (which influences its stability). The lesions with higher percent of diameter stenosis often cause clinically overt ischemia, however, many precursors of culprit lesions in acute coronary syndromes are nonobstructive [22]. Plaque characteristics that are associated with the risk of plaque rupture include microcalcification, positive remodeling, large necrotic core, and thin fibrous cap [23]. The equivalents of those features described for coronary CT include the following: spotty calcification, “napkin ring” sign, positive remodeling and low attenuation plaque [24]. Those features are known to predict future major adverse cardiovascular events [25]. In a large, multicenter, prospective study of coronary CT (SCOT-HEART—Scottish COmputed Tomography of the HEART), it was demonstrated that the above mentioned vulnerable coronary plaque characteristics are associated with a tripling of the risk of coronary heart disease death or nonfatal myocardial infarction [26]. Spotty calcification in this study was defined as focal calcification within the coronary artery wall < 3 mm in maximum diameter [27]. In another multicenter, case–control study of stable patients without diagnosed coronary artery disease (ICONIC—Incident COroNary Syndromes Identified by Computed Tomography) it was also demonstrated that features of vulnerable (high-risk) plaque (with spotty calcification among them) identified patients at higher risk of experiencing acute coronary syndrome [22]. On the contrary, only 52% of patients in the study and only 31% of coronary lesions that later triggered clinical events demonstrated high-risk plaque features, which indicates that the relationship between vulnerable plaque features in CT imaging and acute coronary syndromes in the future is not extremely close [28].

Benefits of performing coronary CT routinely before percutaneous coronary interventions include the possibility to plan certain aspects of the procedure before it starts. Apart from assessment of the plaque characteristics described above, it is also feasible to identify lesion length and to optimize fluoroscopic viewing angles. Procedure time, contrast volume and radiation dose can potentially be reduced with the use of pre-procedural CT [29], which is especially important in complex procedures performed in heavily calcified lesions. An ongoing P4 trial (Precise Procedural and PCI Plan, NCT05253677) is a randomized study with a non-inferiority design of patients that aims to compare the clinical outcomes of two strategies to guide coronary intervention: CT-guided PCI strategy (investigational technology) and IVUS-guided PCI strategy (comparator). Moreover, noninvasive measurement of fractional flow reserve (considered a gold standard to assess hemodynamic significance of the lesion) is now possible with CT. It utilizes computational fluid dynamics and does not require hyperemia [30]. In heavily calcified lesions it is, however, of very limited use.

### 2.2. Coronary Angiography

CAC areas are seen as radiopaque opacities in the arterial wall which can be visualized on a stable picture or during cardiac motion. Calcification is visible before the administration of contrast media. The presence of calcification is one of the criteria included in the SYNTAX score (which indicates the complexity of the lesion); the densities in arterial walls have to be visible in more than one projection, surrounding the complete lumen of the coronary artery at the site of the lesion in order to fulfill that criterion [31]. It is also worth noting that the presence of CAC on coronary angiography does not automatically indicate hemodynamically significant luminal obstruction. Due to the fact that the sensitivity of coronary angiography to detect CAC and intra- and interobserver reproducibility are not high enough, the use of intravascular imaging is highly encouraged to avoid underestimation of calcification [21]. Figure 2 shows coronary calcifications in different imaging modalities (coronary angiography, intravascular ultrasound, and optical coherence tomography).

### 2.3. Intravascular Imaging: IVUS (Intravascular Ultrasound) and OCT (Optical Coherence Tomography)

Both of these methods are catheter-based and use a probe introduced over a guidewire in a coronary artery. IVUS uses ultrasound waves to create cross-sectional images of the blood vessels (without a need to inject contrast); calcification is visible as a hyperechogenic structure (brighter than the adventitia) with characteristic “shadow” that CAC casts over the deeper structures of the vessel. The ultrasound signal is attenuated by the calcium, which hinders the quantification of calcium thickness behind the leading edge [21]. Parameters that can be quantified are the length of the calcified segment and the size of the circumferential arc (the extent to which the circumference of the vessel is affected by calcium expressed in radial degrees). Also, IVUS can indicate if the calcification is nodular, superficial, or deep. IVUS-detected calcification has been proved to predict stent underexpansion [32].

An improvement over standard IVUS is high-definition IVUS (HD-IVUS, utilizing 60 MHz technology instead of a standard 20–40 MHz) that provides better spatial resolution (approximately 40–60 μm compared to standard 50–200 μm). A rarely used imaging modality is VH-IVUS (virtual histology) in which a colorized tissue map of plaque composition derived from ultrasound images is created, showing different tissue types (fibrous, fibro-fatty, calcified, necrotic core).

OCT uses infrared light to create images of the arterial wall. The resolution is superior to IVUS, but the trade-off is the lower tissue penetration. Due to lower attenuation by calcium compared to IVUS, the full extent of calcific plaque can be visualized and the thickness of it can be assessed (except very thick calcified plaques in which the far side of the plaque can no longer be visualized). OCT has also been shown to predict stent underexpansion [33]. Contrast administration is required during image acquisition. Contrast media leads to blood flushing out of the vessel in order to prevent artifacts in the image. OCT capability to visualize microcalcifications is much better compared to IVUS because of the higher resolution. Lower penetration, however, limits the estimation of deep calcium. The assessment of calcification in lesions with an overlying thrombus or macrophage/necrotic core may be limited. Calcified areas that are present within or behind a necrotic core close to media or within macrophage-rich regions can be missed [34].

Intravascular imaging in PCI procedures in heavily calcified lesions should be liberally used both in the initial stage of the PCI (assessment of calcium severity arc, length, thickness, calcified nodules, selection of most appropriate tools and techniques), after use of lesion-preparation techniques (assessment of calcium debulking, luminal gain and calcium fractures/fissures, stent sizing, landing zones of the stent) and finally after stent implantation (assessment of stent apposition and expansion, potential complications, e.g., dissection, hematoma or thrombus). An inability to cross the lesion with an imaging catheter often indicates that plaque modification techniques should be used upfront [21]. The use of intravascular imaging techniques in procedures performed in lesions with severe CAC enables the attainment of better procedural and clinical outcomes as compared to procedures guided via angiography only, which has been widely demonstrated [32,35].

The main differences and potential advantages and drawbacks of both IVUS and OCT have been summarized in Table 1.

## 3. Plaque Modification Techniques in Calcified Lesions

The PCI procedures in patients with CAC have a high risk of complications such as vessel perforation or rupture. Both short- and long-term outcomes are also impaired, mainly due to stent underexpansion, which potentially leads to in-stent restenosis or stent thrombosis. Thus, in many cases, the use of lesion-preparation techniques is required. The objective of these methods is to modify the plaque to facilitate stent implantation. The desired effects are calcium debulking (reduction in the calcium burden in the lesion) and creation of calcium fractures (which is associated with improved stent expansion) [33]. Techniques that are routinely used include:Cutting balloons and scoring balloons. A cutting balloon is a non-compliant balloon (not expandable over their nominal diameter with overinflation) with microblades (three or four) placed on its surface longitudinally, cutting the plaque [36]. Scoring balloons can be either semi-compliant or non-compliant depending on the manufacturer and model and have scoring elements on its surface (helical elements, cutting the plaque with minimal risk of balloon displacement). The presence of these elements enables effective dilation at lower inflation pressures, which decrease the risk of dissection.High- and super-high-pressure balloons (HPB and SHP). The expansion profile of high-pressure balloons compared to semi-compliant balloons is more uniform and limited which prevents the dogboning effect (a situation in which there is both under- and overexpansion of the balloon which can lead to vessel dissection or perforation) [21]. A super-high-pressure balloon (OPN, SIS Medical AG) has a twin-layer construction that enables inflations up to 35 atm (in selected cases even up to 40–50 atm) with a low risk of balloon rupture [37]. The limitation of these balloons is their stiffness that can make advancing them across the lesion difficult.Intravascular lithotripsy (IVL). The Shockwave Medical coronary IVL system is a fluid-filled balloon angioplasty catheter with two lithotripsy emitters incorporated into the shaft of the 12 mm long balloon (which is 0.014″ guidewire compatible). The balloon is delivered across the lesion and then expanded to 4 atm to enable the energy transfer. The fluid inside the balloon is vaporized using an electrical discharge from the emitters, which as a result creates a rapidly expanding and collapsing bubble that generates sonic pressure waves. The shockwaves pass through the balloon and into the calcified plaque, causing it to crack or fracture. The shockwaves from the lithotripsy emitters disrupt the structure of the calcified plaque without causing significant trauma to the surrounding healthy tissue. The goal is to modify the plaque, making the stent implantation easier. The system comes in 2.5, 3.0, 3.5, and 4.0 mm balloon diameters and should be sized 1:1 to the reference vessel diameter. When the balloon is positioned and expanded to 4 atmospheres, the cycle of 10 IVL pulses is delivered (which is followed by optional brief inflation to 6 atm). Up to 80 pulses per balloon or 120 with the latest generation Shockwave C2+ system can be delivered, with deflation between cycles to allow distal perfusion. The potential drawback of this technique is the bulkiness of the balloon which may hinder the delivery across the lesion. The Disrupt CAD studies demonstrated high procedural success and low 30 day major adverse cardiac event rate [38].Atherectomy techniques: rotational atherectomy (RA, rotablation) and orbital atherectomy (OA). The principle of these techniques is to ablate the calcific plaque while also creating plaque fractures and fissures.Rotational atherectomy was first described in 1987. It employs an olive-shaped burr with diamond chips embedded into it. The rotational atherectomy system consists of a console (which regulates the flow of air to the advancer, controlling burr rotation speed and also displays procedural parameters such as burr speed, duration of atherectomy, decelerations which are sudden drops in rotational speed), an advancer (which is used to control movement of the burr) and the burr itself which is introduced to the coronary vessel over a dedicated guidewire. The burr rotates at high speeds (140,000 to 180,000 rpm) and ablates the calcified tissue in a mechanism known as differential cutting (which preferentially ablates the inelastic tissue without damaging the vessel). The operator should use a pecking motion of the burr, which is a quick back and forth movement of the burr to the lesion and back. Decelerations in burr rotational speed should be avoided in order to prevent the burr stall which is serious complication (a situation in which the burr is stuck in the lesion and can no longer be moved or rotated). The particles of debris are <5 μm in diameter and can pass to the systemic circulation without causing distal embolization, however, some considerations are important to prevent the no-flow or slow-flow phenomenon (short burr runs, pauses between runs, appropriate rotational speed, avoidance of decelerations, appropriate pharmacotherapy, i.e., verapamil, nitrates, proper periprocedural anticoagulation). The burrs come in different sizes (from 1.25 mm up to 2.5 mm) to make it suitable for different vessel diameters (burr size should be <0.7 of a reference vessel diameter). PREPARE-CALC (patients were randomized to a lesion modification with use of either RA or cutting/scoring balloons) and ROTAXUS (randomization to groups with RA followed by stenting or stenting without RA) trials indicated that RA before stent implantation is feasible and effective nearly in all patients with heavily calcified lesions [39,40]. A schematic representation of the RA burr is shown in Figure 3.Orbital atherectomy uses two physical mechanisms: differential sanding and centrifugal forces. The crown rotates eccentrically (off-center), creating an “orbital” motion as it spins around the catheter shaft. Compared to RA, there is only one size of the crown, but use of different speed settings makes it suitable for different vessel sizes (the range of the orbital motion is higher with higher speeds). Two speed settings are available; low speed (80,000 rpm) is often used for the first pass, while the higher speed (120,000 rpm) can be utilized in certain lesions, especially in vessels with a larger diameter. Contrary to RA, OA works bidirectionally both when it is advanced and retracted. The ORBIT II trial has proved the safety of the procedure by indicating a low rate of adverse ischemic events [41]. A large, randomized ECLIPSE trial (OA vessel preparation compared with high-pressure balloons angioplasty and/or cutting balloons) is ongoing [42].Excimer laser coronary angioplasty (ELCA) is a relatively rarely-used method. Photoablation (a process based on the emission of monochromatic coherent light in the ultraviolet range) involves the breaking of molecular bonds within the plaque material, causing it to vaporize or ablate. ELCA catheters also have different sizes available and the ratio between the catheter size and the reference vessel diameter should be 0.5–0.6. The clinical use of ELCA is limited. It is mainly used to tackle lesions that are non-dilatable with conventional methods [43]. In calcified lesions it can be used either as a standalone method or after failed RA. ELCA is the only possible option if the lesion is uncrossable with a microcatheter or a guidewire (such as in chronic total occlusions). A specific application of this method is within underexpanded stents (however, the indication is off-label) [21]. One study of 81 cases confirmed ELCA to be superior to predilation with high-pressure balloon in patients with stent restenosis due to stent underexpansion (OCT confirmed cracks in calcium behind the struts achieved using ELCA) [44].

The invasive approach to tackle heavily calcified lesions is summarized in Figure 4.

Optimal tool selection is a complex issue and depends also on the availability in the cath lab and on the operator’s experience and preferences. If the lesion is crossable with a balloon, it should be predilated and then assessed through intravascular imaging, with further management depending on the fact if there are features of high calcium burden in IVUS/OCT. If advancing the balloon across the lesion is not feasible, a microcatheter can be used and RA or OA can be performed (direct wiring of the vessel with dedicated RA/OA wires is also an option). In the case of a lesion not crossable using a wire upfront, the use of ELCA can be considered. If the balloons do not properly expand after use of lesion modification techniques, IVL may be used on top of them. Final postdilation is performed with NC or SHP balloons in most cases. The above described algorithm proposed by De Maria et al. [45] may be helpful in decision making but certain clinical scenarios may differ from it.

Differences between plaque modification modalities have been presented in Table 2.

An example of a relatively rarely-seen situation is presented in Figure 5. A patient with a stent implanted in proximal segment of right coronary artery (RCA) approximately 6 months prior to admission underwent elective coronary angiography which revealed heavily calcified lesions distally to the previously implanted stent. A strategy that was chosen was to perform RA in distal RCA via guide extension (used as a “protection” of the proximal stent). Finally, a stent was implanted under IVUS guidance with an optimal result.

## 4. Complications

Procedures performed in heavily calcified lesions are more complex and carry higher risks of complications. Each step of the procedure (wiring of the vessel, balloon and stent delivery, stent deployment and expansion) is more difficult and cumbersome in this group of patients and has a higher risk of complications. Moreover, stent underexpansion which is more common in cases with severe CAC, is a strong predictor of restenosis and stent thrombosis. Possible complications include:-Entrapped equipment. There is no universal, straightforward technique to retrieve retained equipment due to a large number of possible scenarios. In the case of an entrapped wire or stent it is possible, for example, to pull them out with a special snare, drag them into the guidewire or guide extension with a balloon or, in some cases, pin them to the vessel wall with a stent. A technique with two or three coronary guidewires entangling the entrapped stent to retrieve it is also feasible in some situations [46]. A specific and especially dangerous situation is the entrapment of the RA burr (burr stall); thus, it is important to avoid decelerations (a sudden drop in rotational speed) during RA. A rare scenario is when all methods to retrieve entrapped equipment fail and cardiac surgery is needed.-No-reflow. The mechanism of this phenomenon is not fully understood (endothelial dysfunction, microvascular obstruction, arteriolar spasm, or distal embolization with microparticles may contribute). It is most common in ST-segment–elevation myocardial infarction and procedures performed in degenerated vein grafts, but also after the use of atherectomy devices [47]. Various pharmacological agents such as adenosine, nitroprusside, nicardipine, and verapamil may be administered. Verapamil, heparine and nitroglycerine are commonly used in flush solutions during RA [48].-Dissection. It is essential to promptly identify and to properly manage coronary dissection, since major dissections may potentially result in the obstruction of coronary flow and hemodynamic collapse. Establishing or maintaining blood flow in the artery with dissection is crucial. In most cases, a dissection needs to be covered with a stent. If intravascular imaging is used, IVUS is preferable over OCT. During OCT passage an antegrade injection of contrast is needed, which may further aggravate a dissection or cause a new intimal tear.-Perforation of the vessel. Perforations are generally rare, however, the presence of CAC and use of atheroablative devices or cutting balloons are well-known predictors of perforation [49]. The strategy of management is determined by the location and severity of coronary perforation. Most commonly, prolonged balloon inflation at the site of perforation is performed. If it fails, a covered stent must be implanted. In distal perforations coil embolization is possible. Transthoracic echocardiography has to be completed to assess for cardiac tamponade and need of pericardiocentesis.

In patients with severe CAC, in which a stent has been implanted without proper calcium debulking, stent underexpansion is a relatively common problem and is challenging to manage in this setting. IVL and ELCA can be used, however, they are still considered as off-label. The international, multicenter CRUNCH registry included 70 patients with stent underexpansion due to heavy underlying calcification treated with IVL (with a median time of 49 days from stent implantation to IVL therapy) [50]. No IVL-related procedural complications or major adverse cardiovascular events were observed. Coronary lithotripsy, used to increase lumen and stent dimensions in underexpanded stents, secondary to heavily calcified lesions, can be considered safe and effective (however, further research is needed).

A less common strategy, performed, for example, in cases of late restenosis with calcified neoatherosclerosis inside the stent, is a so-called “stentablation”; RA performed across previously implanted stent. RA can ablate the calcified intrastent tissue, which allows vessel dilation and repeat stent implantation with adequate expansion [51]. Extensive data are missing, but case reports and case series are promising. An ongoing ROTA-ISR randomized trial (RA with subsequent balloon angioplasty and drug-coated balloon vs. balloon angioplasty and drug-coated balloon without RA) will reveal the role of debulking with RA for the treatment of in-stent restenosis with neointimal hyperplasia.

## 5. Long-Term Management and Antiplatelet Treatment

The duration of antiplatelet therapy (dual antiplatelet therapy (DAPT) with aspirin and P2Y12 inhibitor or “triple” therapy in patients requiring oral anticoagulant) can be prolonged in selected patients over a standard 6 months (in chronic coronary syndromes) or 12 months (in acute coronary syndromes). Complex PCI factors determining high thrombotic risk (which are factors in favor of lengthening combination antiplatelet therapy) are usually defined as: implantation of three or more stents, treatment of three or more lesions, total stent length > 60 mm, revascularization of left main coronary artery, bifurcation stenting with an implantation of two or more stents, chronic total occlusion, stenting of the last remaining vessel and stent thrombosis on antiplatelet treatment in history. There are also studies in which rotational atherectomy use for severely calcified lesions is a part of the definition of complex PCI [52,53]. Specific clinical scales may be used to guide the duration of antiplatelet therapy (applied on an individualized basis with clinical context taken into consideration). DAPT score assesses thrombotic risk and predicts which patients will benefit from prolonged DAPT after PCI. PRECISE-DAPT assesses bleeding risk and helps to identify patients in whom antiplatelet therapy should be shortened. Patients undergoing PCI procedures performed in heavily calcified lesions should receive optimal anticoagulation and antiplatelet therapy according to standard practices (in many cases DAPT may be prolonged due to complexity of the procedures). Data considering the potential use of more potent P2Y12 inhibitors (ticagrelor, prasugrel) instead of clopidogrel as a standard practice in CAC patients undergoing elective PCIs are scarce. The TIRATROP study compared clopidogrel-based strategy to ticagrelor-based strategy in patients treated with rotational atherectomy (with primary endpoint based on troponin release within the first 24 h). The study failed to demonstrate the superiority of ticagrelor over clopidogrel to limit the extent of myocardial injury during RA procedures [54]. Glycoprotein IIb/IIIa inhibitors (such as abciximab) have been previously shown to reduce creatine kinase-myocardial band release and slow-flow/no-reflow phenomenon with RA in studies published in the early 2000s; however, they are no longer recommended for routine use (they may be considered in selected cases for which high thrombotic risk outweighs bleeding risk) [55,56].

## 6. Ongoing Research and Future Perspectives

There are several ongoing clinical trials in which patients with CAC treated with dedicated techniques are being enrolled [57].

The ISAR-CALC 2 (Comparison of Strategies to Prepare Severely Calcified Coronary Lesions; NCT05072730) trial randomizes patients with severely calcified, undilatable lesions to either super-high-pressure balloon strategy or IVL strategy. The primary endpoint will be final angiographic minimal lumen diameter (MLD) after stent implantation [58].

The ROTACUT (Rotational atherectomy combined with Cutting balloon to optimize stent expansion in calcified lesions; NCT04865588) trial compares RA plus cutting balloon with RA plus plain conventional angioplasty. Short-term periprocedural outcomes, as well as minimal stent area, stent apposition, stent expansion (assessed with IVUS) are being evaluated.

PREPARE-CALC COMBO (Evaluation of a Strategy to Prepare severely Calcified Coronary Lesions with a Combination of rotational atherectomy and Modified Balloons Trial; NCT04014595) compares patients treated with RA plus cutting balloon strategy to historical data from the randomized PREPARE-CALC trial (patients underwent lesion preparation with cutting/scoring balloons or RA alone). Primary endpoints include in-stent acute lumen gain using quantitative angiographic analysis and stent expansion using OCT.

The ECLIPSE (Evaluation of Treatment Strategies for Severe Calcific Coronary Arteries: OA vs. Conventional Angioplasty Technique Prior to Implantation of Drug-Eluting Stents; NCT03108456) randomized controlled trial compares plaque modification with OA vs. conventional balloon angioplasty. CROWN (Calcium Reduction by Orbital Atherectomy in Western Europe; NCT06035783), another study dedicated to OA, evaluates minimal stent area with OCT after OA.

Ongoing studies on IVL include: the Short-Cut (Shockwave Lithoplasty Compared to Cutting Balloon Treatment in Calcified Coronary Disease Trial; NCT06089135) trial (patients randomized to IVL vs. cutting balloon in two cohorts—those prepared with or without RA), the DECALCIFY (Prospective, Randomized, Controlled, Multicenter Study for the Treatment of Calcified Coronary Artery Lesions With Rotational Atherectomy vs. Intravascular LithotripsY; NCT04960319) trial (randomization to IVL vs. RA; in-hospital major adverse cardiovascular events and OCT-evaluated stent expansion as endpoints), the SONAR (Shockwave Balloon or Atherectomy With Rotablation in Calcified Coronary Artery Lesions; NCT05208749) trial (IVL vs. RA, assessment of postprocedural myocardial infarction), and the VICTORY (Value of IVL Compared to OPN Non-compliant Balloons for Treatment of Refractory Coronary Lesions; NCT05346068) trial aiming to evaluate final stent expansion following a strategy of lesion preparation with either IVL or a super-high-pressure balloon in patients with heavily calcified coronary lesions undergoing coronary stent implantation.

Studies evaluating the safety and efficacy of innovative devices that are underway include the ACTIVE study (Safety and Efficacy Study of the SoundBite Crossing System With ACTIVE Wire in Coronary CTOs; NCT03521804) that assesses a novel coronary SoundBite Crossing System in a subject population with chronic coronary artery disease including chronic total occlusion. The guidewire that is used in the study penetrates calcium using pressure pulses characterized by high amplitude, rapid rise time, and short duration. Another novel device is a T-wave IVL catheter system (Suzhou Zhonghui Medical Technology; NCT05552131).

There are also ongoing studies comparing multiple plaque modification modalities. The ROLLING-STONE study (IVL and/or Mechanical Debulking for Severely Calcified Coronary Artery Lesions; NCT05016726) evaluates the intra-procedural and long-term effects of IVL and/or non-balloon mechanical debulking devices, prior to and/or after coronary stenting in patients with complex calcified coronary artery lesions. Finally, the ROLLERCOASTR randomized trial (Rotational Atherectomy, Lithotripsy, or Laser for the Treatment of Calcified Stenosis; NCT04181268) will compare the safety and efficacy of RA, IVL and ELCA.

## 7. Conclusions

CAC is a common problem typically present in older individuals. In patients with certain risk factors (e.g., diabetes or chronic kidney disease) the formation of coronary calcification can be more rapid and can lead to clinically overt symptoms in younger age. It is associated with a higher rate of cardiovascular events and is a factor affecting every step of PCI procedures (crossing the lesion with a guidewire, balloon delivery and inflation, stent delivery and expansion). The above mentioned lead to suboptimal PCI results and higher rate of complications. Careful patient selection and pre-procedural planning (CT scan with assessment of CAC extent) is of special importance. Intravascular imaging should be routinely used during PCI procedures in lesions with extensive calcification in order to choose the most appropriate tools and strategy and to assess the final result of the procedure. In many patients with CAC, one of the plaque modification techniques have to be performed prior to stenting, in order to facilitate stent implantation and to prevent stent underexpansion and other possible complications.

## Figures and Tables

**Figure 1 healthcare-12-00520-f001:**
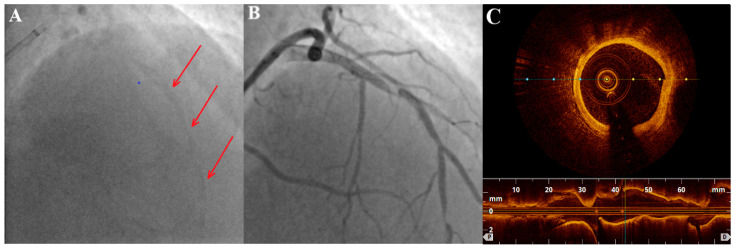
A 33-year-old patient with end-stage chronic kidney disease requiring dialysis. (**A**) massive calcification (marked by arrows) in left anterior descending (LAD) artery before contrast administration; (**B**) angiography of LAD revealing significant stenoses; (**C**) OCT cross-section image of LAD with circumferential calcification.

**Figure 2 healthcare-12-00520-f002:**
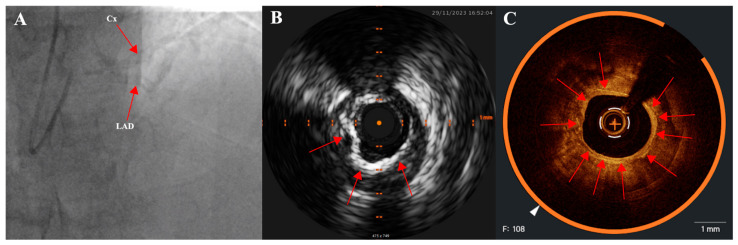
Coronary calcification visualized using different imaging modalities: (**A**) coronary angiography; (**B**) intravascular ultrasound; (**C**) optical coherence tomography (circumferential calcium, 360° arch). Arrows are pointing at calcium deposits.

**Figure 3 healthcare-12-00520-f003:**
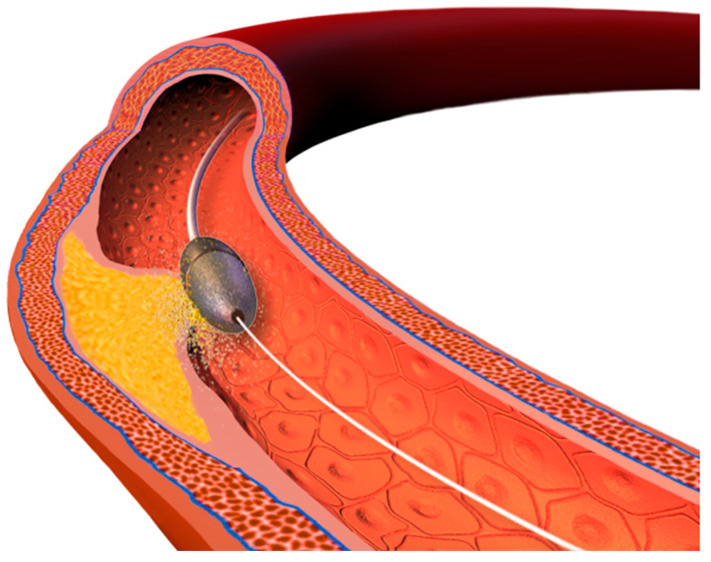
Rotational atherectomy, schematic representation.

**Figure 4 healthcare-12-00520-f004:**
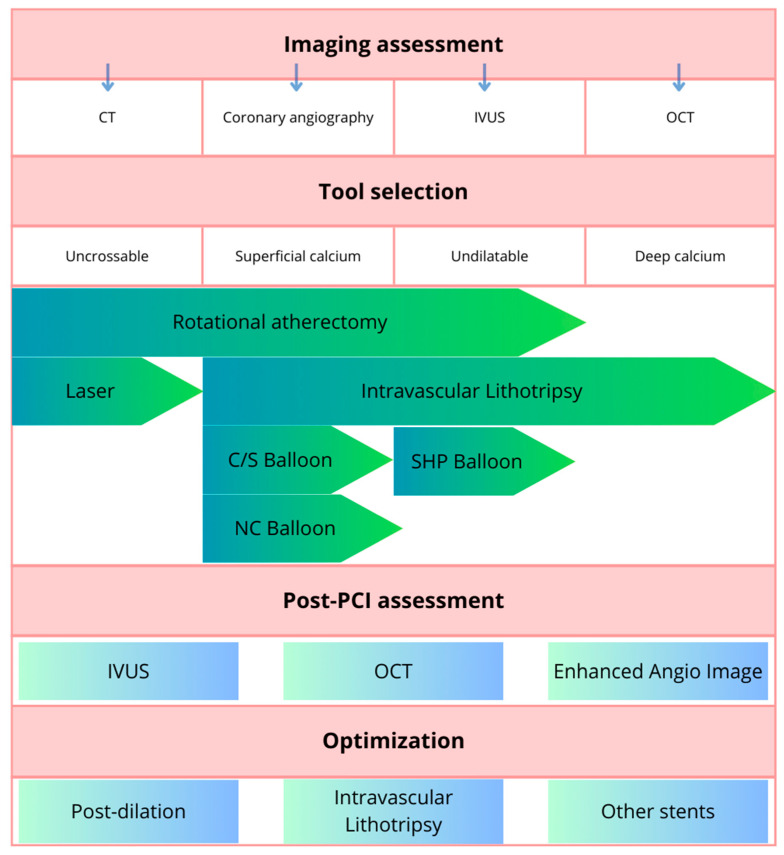
Summary of recommendations for invasive approach in heavily calcified lesions. Adapted from [21]. Abbreviations: CT, computed tomography; IVUS, intravascular ultrasound; OCT, optical coherence tomography; NC, non-compliant; C/S, cutting/scoring; SHP, super-high-pressure; PCI, percutaneous coronary intervention; IVL, intravascular lithotripsy.

**Figure 5 healthcare-12-00520-f005:**
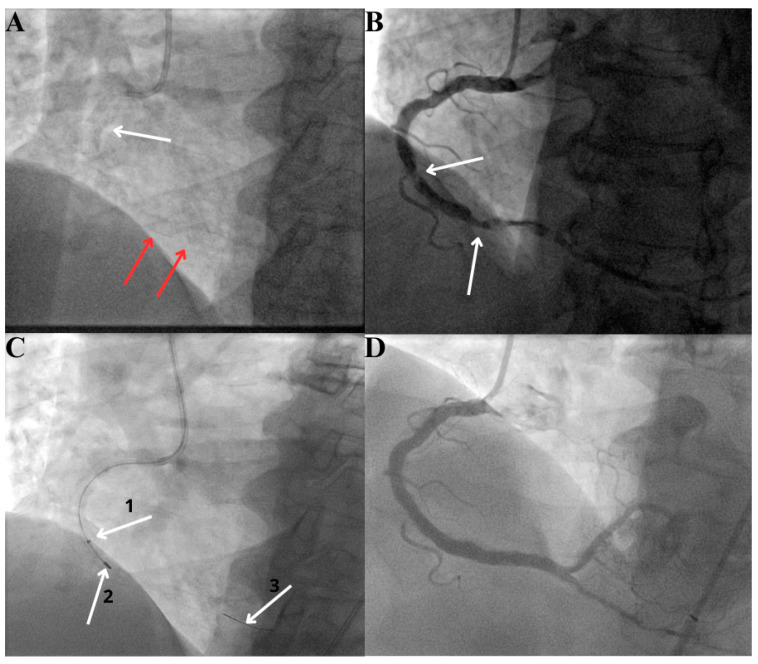
(**A**) Stent in proximal right coronary artery (RCA, stent marked by white arrow) and massive calcification in distal RCA (red arrows); (**B**) significant lesions distally to previously implanted stent (arrows); (**C**) 1. distal tip of the guide extension, 2. rotational atherectomy burr, 3. RotaWire Floppy (**D**) final angiography.

**Table 1 healthcare-12-00520-t001:** Comparison between intravascular ultrasound (IVUS) and optical coherence tomography (OCT). CAC—coronary artery calcification. “-”: least accurate; “+++”: most accurate.

	IVUS	OCT
Spatial resolution	50–200 μm	15–20 μm
Need of contrast injection	No	Yes
Duration of data aquisition	2–4 min	<10 s
Visualization of severe CAC	+++	+++
Visualization of mild/moderate CAC	++	+++
Deep calcium	+++	++
Calcium arch	+++	+++
Calcium thickness	-	+++
Longitudinal calcium length	+	+++
Non-homogenous plaque/necrotic core	+++	+

**Table 2 healthcare-12-00520-t002:** Comparison between different plaque-modification techniques.

	Super-High Pressure Ballons	Cutting Balloons	Scoring Ballons	RA	OA	IVL	ELCA
Physical mechanism	High pressure with uniform expansion	Radial incision	Focal force/slice	Differential cutting	Differential sanding and pulsatile forces	Pulsatile mechanical energy	Photoablation (vaporization)
Guidewire	0.014″	0.014″	0.014″	0.009″ (Rotawire Floppy/ Extra support)	0.012″/0.014″ (Viperwire)	0.014″	0.014″
Pressure	Up to 45 atm	Rated burst pressure 12 atm	Rated burst pressure 16 atm			Typically 4–6 atm	
Ablation direction				unidirectional	bidirectional		
Rotational speed				140,000–180,000 rpm	80,000/120,000 rpm		
Particle size				5–10 μm	<2 μm		
Specific considerations/applications				- balloon uncrossable lesions - late in-stent restenosis with calcified neoatherosclerosis	- long diffuse calcification - large diameter vessels - aorto-ostial lesion - more than one lesion requiring modification, with a difference in reference vessel diameter	- focal lesion - bifurcation with both branches requiring calcium modification - stent underexpansion with calcium outside the stent (off-label)	- uncrossable lesions - thrombotic calcified lesions - stent underexpansion with calcium outside the stent (off-label)

## Data Availability

No new data were created or analyzed in this study. Data sharing is not applicable to this article.

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
