# Peer review of "The Significance of Coronary Artery Calcification for Percutaneous Coronary Interventions"

_healthcare, 2024, doi:10.3390/healthcare12050520_

Round 1
Reviewer 1 Report
Comments and Suggestions for Authors
Lis and colleagues provide an extensive review of the significance of coronary artery calcification for PCI – topic of high clinical interest in everyday practice. Generally, the manuscript is up-to-date and provides basic information about pathophysiology, significance and principles of severely calcified lesions treatment. However, I believe that procedural part of the manuscript not sufficiently describe the topic.
1. Severely-calcified lesion PCI is a high-risk procedure. I suggest additional paragraph regarding potential complications and its management.
2. What are pros, cons, and contraindications for each method of plaque modification techniques? I suggest adding a table, summarizing most important information regarding each technique, similar to Table 1.
3. I suggest adding paragraph regarding post-PCI anti-thrombotic treatment.
4. Coronary calcifications may cause stent under-expansion. May the authors add short paragraph regarding treatment of stent underexpansion?
5. Authors remark potential CT-based fractional flow reserve assessment. Do the Authors believe that such diagnostic tool has potential clinical utility in highly-calcified lesions?
6. I suggest adding a paragraph regarding future perspectives, ongoing studies, etc.
7. I believe that authors should review the manuscript language-wise.
8. If possible, I suggest adding central figure/graphical abstract.
Comments on the Quality of English Language
Stylistic corrections should be applied.
Author Response
Dear Reviewer,
We appreciate you for your time in reviewing our paper and providing valuable and insightful comments. The authors have carefully considered the comments and tried our best to address them. The authors welcome further constructive comments if any.
Below we provide the point-by-point responses. All modifications in the manuscript (added content) have been highlighted in green.
- A separate section addressing complications has been added (Section 4., line 371)
- A table has been added.
- A separate section has been added (Section 5., line 427)
- Stent underexpansion has been discussed in Section 4.- complications, line 409
- The statement regarding FFR-CT has been corrected (line 166)
- A separate section has been added (Section 6., line 456). Additionally, some of the ongoing studies have been discussed in paragraph describing CT (line 160) and in section regarding complications (line 423)
- Minor corrections have been made.
- Central illustration has been added.
Best regards,
Paweł Lis
Reviewer 2 Report
Comments and Suggestions for Authors
To the Authors
This manuscript gives a well-written up-to-date overview on the options for treatment of calcified coronary lesions.
We only have a few minor remarks
l.98 : that suggests
l. 125-127: It is probably not correct to state the rate of acute coronary syndromes increases with higher percent of diameter stenosis. Also in the reference cited by the authors, 65% of patients who presented with an ACS had nonobstructive lesions. It is well known that stable coronary plaques (with a well developed fibrotic cap) lead to the most significant stenoses.
We feel that the authors should modify this statement
Fig 1: it is unclear at what level the IVUS and OCT images ware taken, LAD or Cx - same lesion ? panel A shows severe calcification of both LAD and Cx. Please clarify.
l 290: mechanisms
l 340: Fig 5 should be part of the discussion section, not introduce new information or illustration in the summary section.
In our version, author contributions are not specified.
Comments on the Quality of English LanguageThe English language is fine, just a few typo's to be corrected (sie comments to authors)
Author Response
Dear Reviewer,
We appreciate you for your time in reviewing our paper and providing valuable and insightful comments. The authors have carefully considered the comments and tried our best to address them. The authors welcome further constructive comments if any.
Below we provide the point-by-point responses. All modifications in the manuscript (added content) have been highlighted in green.
l. 125-127 (in revised manuscript: 133-135): the statement has been corrected
Fig. 1 (in revised manuscript: Fig. 2): the images are not from the same case. Our aim was simply to present different imaging modalities in this figure.
Fig. 5 has been moved to the introduction section and renamed as Fig. 1.
Additionally, new sections and central illustration have been added according to another reviewer's suggestions.
Best regards,
Paweł Lis
Round 2
Reviewer 1 Report
Comments and Suggestions for Authors
Thank you for the changes in manuscript.
I believe that the paper now thoroughly describes the topic of calcified lesion management.
Best regards.
Comments on the Quality of English LanguageThank you for changes.